# Education for Sustainable Development and Meaningfulness: Evidence from the Questionnaire of Eudaimonic Well-Being from German Students

**DOI:** 10.3390/ijerph19116755

**Published:** 2022-05-31

**Authors:** Johannes Klement, Wiltrud Terlau

**Affiliations:** Department of Applied Sciences, International Center for Sustainable Development, Hochschule Bonn-Rhein-Sieg, 53757 Sankt Augustin, Germany; wiltrud.terlau@h-brs.de

**Keywords:** education for sustainable development, sustainable development goals, eudaimonic well-being, well-being, SDG 3, SDG 4

## Abstract

Education for Sustainable Development (ESD, SDG 4) and human well-being (SDG 3) are among the central subjects of the Sustainable Development Goals (SDGs). In this article, based on the Questionnaire for Eudaimonic Well-Being (QEWB), we investigate to what extent (a) there is a connection between EWB and practical commitment to the SDGs and whether (b) there is a deficit in EWB among young people in general. We also want to use the article to draw attention to the need for further research on the links between human well-being and commitment for sustainable development. A total of 114 students between the ages of 18 and 34, who are either engaged in (extra)curricular activities of sustainable development (28 students) or not (86 students), completed the QEWB. The students were interviewed twice: once regarding their current and their aspired EWB. Our results show that students who are actively engaged in activities for sustainable development report a higher EWB than non-active students. Furthermore, we show that students generally report deficits in EWB and wish for an improvement in their well-being. This especially applies to aspects of EWB related to self-discovery and the sense of meaning in life. Our study suggests that a practice-oriented ESD in particular can have a positive effect on the quality of life of young students and can support them in working on deficits in EWB.

## 1. Introduction

For a decade now, we have been observing a strongly increasing desire among our students to deal with technologies, forms of entrepreneurship, production or lifestyles that have a sustainable and social added value. This is inter alia reflected in the fact that students are increasingly asking about solutions to sustainable development issues and want to contribute specifically to the Sustainable Development Goals (SDGs) [1]. According to our observations, this increasing desire is also accompanied by a change in the idea of well-being or prosperity of young people towards concepts that include aspects of personal fulfilment and personal expression. One such type of well-being has been increasingly defined in recent decades as Eudaimonic Well-Being (EWB), “referring to the quality of life derived from the development of a person’s best potentials and their application in the fulfilment of personal expressive, self-concordant goals” ([2], p. 41).

Both EWB and sustainable development are subject to a development process. They require action competences and are the result of concrete actions. The fourth SDG of “Quality Education” in target 7 calls for “that all learners acquire the knowledge and skills needed to promote sustainable development” [3]. In the field of teaching, such a contribution to sustainable development can happen when we focus on teaching skills that enable young people to identify problems of unsustainable development and put their knowledge of sustainable development into practice [4]. This type of teaching, known as Education for Sustainable Development (ESD), covering aspects of Social Learning [5], was developed from Agenda 21 ([6], p. 36) in order to help young people to shape the environment in which they live in the sense of sustainable development and to develop their best potentials [4]. Education for sustainable development aims at empowering participation and cooperation and requires learning processes that are designed to be participatory as a learning and design process [7]. Target 4 of SDG 3 “Good Health and Well-Being” [8] calls for measures to promote mental health and human well-being. This is where SDGs 3 and 4 intertwine. Through the mechanism of a concept of well-being based on action and personal expressiveness, ESD (SDG 4) contribute to SDG 3 in the practical implementation of the goals of the 2030 Agenda.

A link between the practical implementation of SDGs and well-being can exist on several levels. Practical implementation in terms of pro-social behavior or donation to public goods can lead to an increase in emotional or hedonistic well-being in the sense of a warm glow [9] or helper’s high ([10], p. 42 et sqq.). It may also lead to an increase in social or eudaimonic well-being [11] or life satisfaction [12]. Especially due to the social aspect of volunteering, such behavior also has a positive effect on the mental and physical health [13,14] and self-esteem [15]. If sustainable engagement or ESD specifically include nature-based or outdoor activities, health and well-being can also be increased through the connection to nature [16,17]. Well-being can also be negatively affected; empathy, in particular, can lead to people who help in disaster settings to have their well-being or mental health impaired [18].

Despite extensive research, there is little evidence to date that, firstly, relates to the situation of young people who are still in a phase of personal development and, secondly, offers a detailed analysis of EWB in all its aspects. With reference to our initial observation, we want to focus on the different aspects of EWB and specifically address the question of the extent to which a practice-oriented implementation of SDG 4 within the framework of ESD contributes to an improvement in EWB for young people in the sense of SDG 3. Based on the Questionnaire for Eudaimonic Well-Being (QEWB) [2], we investigate to what extent (a) there is a connection between EWB and practical commitment to the SDGs and whether (b) there is a deficit in EWB among young people in general. We also want to use the article to draw attention to the need for further research on the links between human well-being and commitment for sustainable development. We examine the following two hypotheses:

**Hypothesis** **1** **(H1).**
*Young people who are actively engaged in implementing aspects of the SDGs experience more moments of personal fulfilment and report higher levels of EWB than young people who are not actively engaged.*


**Hypothesis** **2** **(H2).**
*Young people are still in a phase of developing their personality and we expect this to be reflected in their EWB. In such that there is a significant deviation of the actual EWB from the desired EWB.*


### 1.1. Theory

#### 1.1.1. Eudaimonia

According to Aristoteles, a successful way of life is characterized by virtuous action and a happy state of mind that is associated with it. This state, eudaimonia, is usually described as a superior good that is self-sufficient in itself and is the ultimate goal of all action ([19], 1097b). Eudaimonia is not (only) determined by external factors, but arises from good behavior within oneself. For Aristotle, eudaimonia is attainable in four ways ([20], 1361a): through prosperity that comes from virtue; independence of life; a pleasant and safe existence; or a flourishing state of property and body with the ability of preserving and using it. In concrete terms, the individual components of eudaimonia are then many and good friends, wealth, healthy and numerous offspring, a happy old age, plus physical advantages, such as health, beauty, strength, great stature, athletic prowess, reputation, fame, good fortune, virtue (ibid.). Furthermore, according to Aristotle, humans are social and altruistic beings who value community. Eudaimonia is thus only desirable in a social context and friendship and the possibility of doing good to friends are among the highest goods of his ethics ([19], 1169b). Even though eudaimonia is a concept of happiness that places virtues in the foreground, the principle of pleasure is not completely rejected, but is assigned only secondary importance. It is still a part of a happy life ([19], 1153b).

#### 1.1.2. Eudaimonic Well-Being

The idea of translating eudaimonia into a psychological context was particularly formulated by Alan Waterman in the early 1990s as Eudaimonic Well-Being [21,22,23]. The idea behind this concept is to translate one of the foundations of Aristotelian ethics—“human good turns out to be activity of soul accordance with virtue” ([19], 1098a)—into the context of modern positive psychology. Waterman presented the concept of personal expressiveness as an individualistic and subjective account of the objective concept of eudaimonia, arguing that eudaimonia occurs when activities are congruent with a person’s true self and the person is fully engaged. Expanded by Waterman and other authors over time, eudaimonia or EWB is often used synonymously with many aspects of positive psychology, including intrinsic motivation [24], peak experiences [25], creativity, self-compassion, wisdom, autonomy, and elevation [26] or flow experiences [27].

In our study, we are specifically interested in the aspects of EWB that emerge from the personality development of young people and from taking action for sustainable development. A suitable operationalization of these aspects is provided by the Questionnaire for Eudaimonic Well-Being (QEWB) developed by Waterman et al. [2], which is the methodological basis of our study.

#### 1.1.3. Research Based on the Questionnaire for Eudaimonic Well-Being

We use Waterman et al.’s [2] QEWB firstly because the survey differentially captures various aspects of EWB we think are important regarding our context and secondly because it has already been successfully implemented in various international contexts. The QEWB has been translated into and successfully validated in multiple languages, among them Polish [28], Italian [29], Turkish [30], Spanish [31], or Japanese [32]. We could not find a translation into German and we thus offer a translation into German of our own (s. Appendix A). Various authors have shown in different cultural contexts that the QEWB has good psychometric properties with regard to surveying students [2,30,33,34,35], but also adults [29,32,36].

## 2. Methods

### 2.1. Participants

A total of 114 students of the Hochschule Bonn-Rhein-Sieg, University of Applied Sciences in Sankt Augustin, Germany of the ages of between 18 and 34 years, took part in the study in two groups. The first group consisted of 28 students (19 females and 9 males) who were involved in (extra)curricular activities for sustainable development. The second group comprised 86 students (37 females and 49 males) from the Department of Economics and Business Administration, who were not engaged in such activities.

### 2.2. Measures

Questionnaire for Eudaimonic Well-Being (QEWB), developed by Waterman et al. [2], which we translated into German. The translation can be found in the Appendix A. The questionnaire consists of a total of 21 questions covering the following aspects of EWB according to Waterman et al.: (1) self-discovery, (2) perceived development of one’s best potentials, (3) sense of purpose and meaning in life, (4) investment of significant effort in pursuit of excellence, (5) intense involvement in activities, and (6) enjoyment of activities as personally expressive ([2], p. 44). Responses to each of these 21 items are made on a five-step Likert scale ranging from 0 (strongly disagree) to 4 (strongly agree), whereas 7 questions are written in the negative direction and are reverse scored. A total score is calculated ranging from 0 to 84 (85 points); a higher score identifies participants with higher EWB.

### 2.3. Procedure

In the winter term 2018/19, we interviewed students who were committed to sustainable development issues during a hackathon at our university where they presented their work on the SDGs and collaborated in developing projects for local implementation of the SDGs and put them into practice. We also interviewed students who initiated their own projects in their free time: These projects include an upcycling initiative, an urban gardening project, and a student association promoting local sustainability and intercultural exchange. In the following, we refer to this group as Sample 1. In the context of lectures in the Department of Economics and Business Administration, we had students who were not committed to sustainable development fill out the QEWB twice. Firstly, the students filled out the questionnaire in relation to their current life situation (Sample 2a) and secondly regarding their desired life situation (Sample 2b): “What would your answers to the questionnaire be, if you had achieved the life you wanted?”

## 3. Data Analysis

Due to the small size of the two samples (Samples 1 and 2a), we used a Wilcoxon rank-sum test to test whether the distributions of the responses of both groups were identical. The null hypothesis was rejected with a Z-score of −1.86 when alpha < 0.031. We thus assumed that the distributions of both samples were significantly different from each other. Comparing the responses of Sample 2a and 2b, the null hypothesis was also rejected when alpha < 0.031 with a Z-score of −1.86.

In Sample 1, the responses on the 85-step range were between 45 and 74, with a mean = 57.77 with SD = 8.17, median = 60 and mode = 62. Cronbach’s alpha is 0.7. In Sample 2a, the responses were lower overall: The scores ranged between 33 and 71, with a mean = 52.69 with SD = 8.15, median = 53 and mode = 55. Cronbach’s alpha is 0.72. The distributions of both samples are thus slightly right skewed, but quite close to the normal distribution. The same applies to Sample 2b: the values are highest overall and slightly above the values of Sample 1. The range is between 50 and 79, with a mean = 63.25 with SD = 7.4, median = 64 and mode = 72, whereas Cronbach’s alpha is 0.61.

Compared to the original study by Waterman et al. [2] who verified the questionnaire in two groups with students in the United States, the means of Sample 1 (59.7) and Sample 2a (52.8) are slightly above and below the average of 56.8 and 54.6 points of the two groups of American students who were interviewed by Waterman and colleagues, respectively. Schutte, Wissing, & Khumalo [35] report an average score of 58.9 points for South African students.

## 4. Results

Table 1 shows the average score for each question of the QEWB by sample; the standard deviation is given in parentheses.

Regarding Hypothesis 1, we first analyzed the responses of Samples 1 and 2a. As shown in the previous section, Sample 1 students report a 9.6% higher average EWB than Sample 2a (57.77 vs. 52.69). The average score in Sample 1 is 2.75, that in Sample 2a is 2.51. The comparison shows that both distributions are qualitatively very similar, which is also confirmed by a correlation coefficient of 0.87. Thus, the additional EWB of sample 1 seems to be almost evenly distributed over all items.

However, despite the high degree of similarity, there are a few distinct differences. For example, participants in Sample 1 report high EWB especially when it is directly related to actions (see, e.g., items 5, 8, 15, 18, or 19). Students report the highest score for willingness to put effort into things and achieve excellence (item 19). However, also, in Sample 2a the answers in these items are above average. The scores in both samples are rather below average for items that relate more to aspects of self-discovery and finding meaning (see, for example, items 6, 9, 16, or 21). Further noticeable is the drop in scores for items 3 and 10. Whereby it is also not really clear with what justification they are in this questionnaire, since they have not necessarily a relation to the basic idea of eudaimonia nor to the underlying theory of EWB.

Regarding our second hypothesis and the comparison between Sample 2a and Sample 2b in particular (52.69 vs. 63.25, and 2.51 vs. 3.01 on average per item) it is evident that the students of the second sample wish for a significantly higher EWB, which is also higher overall than that of Sample 1. Additionally, both distributions are qualitatively more different, which is also indicated by a correlation coefficient of 0.53. It is also clear that the increase in EWB desired by students is primarily derived from items that relate more to the aspects of self-discovery and finding meaning. In particular, items 2, 6, 9, or 21 show very significant increases in EWB. A total of 45% of the total desired increase in EWB results from these four items. However, students also report desirable increases in items that relate more to actions (cf. items 1, 20, or 21), although the overall increase in this area is not as large.

In order to structure our findings and present them more concretely, we separated the EWB in Table 2 into items that relate more to self-discovery and finding meaning in life and items that relate more to taking action, although the individual items are not always clearly distinguishable. Table 2 therefore serves more to clarify the results and the observed trends than to provide evidence of psychological aspects.

Table 2 highlights the differences discussed. Sample 1 reports a slightly increased well-being on average compared to Sample 2a. Concurrently, the students wish for increased well-being, especially regarding aspects of self-discovery and finding meaning in life.

## 5. Discussion

Our results show that students who engage in sustainable development and ESD report slightly higher EWB. Furthermore, we show that students in general desire a higher EWB, especially concerning aspects of self-discovery and finding meaning in life. Quantitatively, the EWB scores we surveyed are in line with those of students from other countries. e.g., Waterman et al. [2] for students from the USA or Schutte, Wissing, & Khumalo [35] for students from South Africa. Only in comparison to Japanese students, our students perform significantly better ([32], Table 1).

Regarding whether the increased scores of sustainably committed students are a sign of self-selection, or ESD actually increases EWB, we believe there is causality in both directions. Son & Wilson [11] for example show that volunteering has an increasing effect on EWB, regardless on the amount of time invested in such activities. However, people who report a higher EWB are at the same time also more likely to volunteer; and if they do, to volunteer more.

In designing this study, we had in mind the question of how far we can contribute positively to students’ EWB in the context of ESD and social learning, on the one hand, and how we can motivate students to take more sustainability initiatives, on the other. Our results encourage us to take action from two directions: On the one hand, using the classic design competence of ESD and the offer of practical implementation possibilities to implement sustainable solutions [4,38]. On the other hand, it seems to make sense to expand the concept of ESD, especially the design competences, with psychological components, since EWB can also be learned theoretically to a certain extent. Kiaei & Reio [39], for example, show that goal aspiration and metacognition—referring to the knowledge and control over one’s own cognitions—are significant predictors of EWB. Both goal aspiration and metacognition are learnable.

In terms of a stronger connection between SDG 3 and SDG 4/ESD, we should therefore not only focus on aspects of social, economic, or ecological sustainability, but also consider the (eudaimonic) well-being of students and include it in ESD. Such a connection could be the subject of future revisions and adaptations of ESD. After all, aspects such as self-awareness or conscious action also seem to be linked to sustainable action.

In a similar approach, examining the quality of life of students that results from musical sophistication, something that is certainly in the spirit of EWB, Cara et al. [37] report a positive effect on the psychological well-being (anxiety and anger) of students. Overall, there is very little literature that specifically examines theoretical and practical approaches to education in relation to the quality of life of students. For example, the level of educational attainment does not seem to affect well-being in terms of life satisfaction [40], but the way in which education is delivered does seem to have an impact. At least, that is what our results suggest.

As a result of our study, we also emphasize the importance of intensively involved in activities. Ishii et al. [32] make a similar observation. They report that this plays an important role in the context of EWB, especially among young people. It is therefore quite possible that our results are specific to young people and cannot be easily extrapolated to older people. However, it again highlights the importance of the need to also do applied and practical teaching, such as ESD.

Without entering into educational-theoretical details—we leave this to people who are more qualified in the field—we believe, however, that the practical education we outlined will lead to more coherence in teaching: Especially in ESD, there is often a discrepancy between the real world experienced by students and theoretically taught, “better” sustainable solutions and theories. Practical approaches can resolve this discrepancy somewhat.

### 5.1. Limitations of the QEWB in the Context of Actions for Sustainable Development and the Nicomachean Ethics

We also want to briefly discuss the concept of the QEWB. According to Aristotle ([19], 1097b), eudaimonia is “the end of all things achievable in action” and is thus the result of (virtuous) action. Only about half of the items of the QEWB explicitly refer to actions. Aspects such as “self-discovery” or “meaning or purpose in life,” on the other hand, are not necessarily part of eudaimonia, since they are not a prerequisite for or necessarily connected to virtuous action. However, a central point of eudaimonia that is missing in the QEWB—which at the same time is a central aspect of commitment to sustainable development—is the social aspect. For according to Aristotle, friendship is among the highest goods and eudaimonia can only be realized in a social context: “The same conclusion [that happiness is complete] also appears from self-sufficiency. For the complete good seems to be self-sufficient. What we count as self-sufficient is not what suffices for a solitary person by himself, living an isolated life, but what suffices also for parents, children, wife, and, in general for fellow citizens, since a human being is a naturally political animal” ([19], 1097b).

### 5.2. Limitations of the Study

Our study focuses on a small and specific part of human well-being. We limit our investigation to the interpretation of EWB as given by the QEWB [2]. Other aspects or concepts of well-being, such as hedonic, psychological/emotional, or especially social well-being, life satisfaction or subjective well-being, remain unaddressed in this study. We do not claim to examine ESD/SDG 4 in the context of well-being comprehensively nor do we provide a clear distinction from other well-being concepts. However, we hope that our study has provided an impetus for further research in what we believe is a very important area.

Our findings are in line with other broader studies of EWB in college students. Due to the small number of respondents, especially in Sample 1, our results are more likely to be qualitative. For more precise quantitative statements and a representation or generalization of our results, a larger number of participants is necessary. For example, there seem to be differences in the individual subcategories for women and men, but the sample size does not allow us to make more precise statements, including on other possible effects.

Furthermore, our classification also remains sketchy, also due to the limitations of the questionnaire. Self-discovery, for example, is a much broader concept than just the arithmetic mean of a few questions from a survey. However, here, too, it makes sense, and is actually necessary in the sense of ESD to examine individual aspects in detail in future analyses in order to actually improve the concept of ESD.

## 6. Conclusions

By documenting and analyzing the Eudaimonic Well-Being (EWB) of 114 students, using the Questionnaire of Eudaimonic Well-Being (QEWB) [2], we showed that young students seem to have deficits in their EWB, especially regarding aspects of personal development, purpose, or self-knowledge. In contrast to a control group, students who engaged in (extra)curricular activities for sustainable development performed significantly better and reported a significantly higher EWB. Whereby there is causality in both directions. A higher EWB leads to engagement and engagement leads to higher EWB.

In addition to benefiting the social, environmental, or economic sustainability, (hands-on) education for sustainable development can contribute to the personal development and well-being of young people. This aspect has hardly been considered in the conception of education for sustainable development to date. Especially in the sense of a stronger commitment to sustainable development, aspects promoting EWB should be increasingly included in teaching and education. In the context of sustainable development, students should not only learn how to improve the environment, but also their own situation.

After all, as research shows, EWB can also be learned in reverse, on a theoretical level based on psychological concepts, such as goal aspiration or metacognition.

## Figures and Tables

**Table 1 ijerph-19-06755-t001:** Average score for each question of the QEWB [37] by sample. Standard deviation in parentheses.

	Sample 1	Sample 2a	Sample 2b
1. I find I get intensely involved in many of the things I do each day.	2.82 (0.72)	2.52 (0.7)	3.31 (0.58)
2. I believe I have discovered who I really am.	2.78 (1.03)	2.3 (1.05)	3.6 (0.71)
3. I think it would be ideal if things came easily to me in my life. *Reversed*	1.61 (1.45)	1.59 (0.99)	0.83 (1.15)
4. My life is centered around a set of core beliefs that give meaning to my life.	2.75 (1.04)	2.6 (1.09)	3.25 (0.86)
5. It is more important that I really enjoy what I do than that other people are impressed by it.	3.37 (0.6)	2.98 (0.95)	3.61 (0.76)
6. I believe I know what my best potentials are and I try to develop them whenever possible.	2.5 (0.68)	2.24 (0.79)	3.62 (0.72)
7. Other people usually know better what would be good for me to do than I know myself. *Reversed*	2.75 (0.97)	3.13 (0.96)	3.29 (1.38)
8. I feel best when I’m doing something worth investing a great deal of effort in.	3.18 (0.72)	2.79 (0.97)	3.06 (1.04)
9. I can say that I have found my purpose in life.	2.29 (1.28)	1.65 (1.02)	3.46 (0.76)
10. If I did not find what I was doing rewarding for me, I do not think I could continue doing it.	1.96 (1.89)	1.87 (1.11)	1.78 (1.35)
11. As yet, I’ve not figured out what to do with my life. *Reversed*	2.53 (1.39)	2.3 (1.05)	2.42 (1.6)
12. I can’t understand why some people want to work so hard on the things that they do. *Reversed*	3.19 (0.98)	3.08 (1.15)	3.04 (1.32)
13. I believe it is important to know how what I’m doing fits with purposes worth pursuing.	3.04 (1.07)	2.74 (0.96)	2.96 (1.11)
14. I usually know what I should do because some actions just feel right to me.	2.78 (0.84)	2.8 (0.89)	3.46 (0.72)
15. When I engage in activities that involve my best potentials, I have this sense of really being alive.	3.44 (0.54)	3.05 (0.85)	3.21 (0.93)
16. I am confused about what my talents really are. *Reversed*	2.29 (1.7)	2.16 (1.08)	2.35 (1.7)
17. I find a lot of the things I do are personally expressive for me.	2.55 (0.84)	2.65 (1.06)	3.15 (1.03)
18. It is important to me that I feel fulfilled by the activities that I engage in.	3.08 (0.87)	2.69 (1.07)	3.41 (0.78)
19. If something is really difficult, it probably isn’t worth doing. *Reversed*	3.6 (0.31)	2.98 (0.97)	3.07 (1.25)
20. I find it hard to get really invested in the things that I do. *Reversed*	2.74 (1.23)	2.34 (1.13)	2.89 (1.34)
21. I believe I know what I was meant to do in life.	2.52 (1.14)	2.23 (1.12)	3.48 (0.84)

**Table 2 ijerph-19-06755-t002:** Differentiating between items of the QEWB rather related to self-discovery and meaning in life vs items rather related to taking action.

	Sample 1	Sample 2a	Sample 2b
EWB related to self-discovery and meaning in life	28.19	26.00	33.67
EWB related to taking action	29.58	26.69	29.58
Total	57.77	52.69	63.25

## Data Availability

The data are available from the authors on request.

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
