# Peer review of "Education for Sustainable Development and Meaningfulness: Evidence from the Questionnaire of Eudaimonic Well-Being from German Students"

_ijerph, 2022, doi:10.3390/ijerph19116755_

Round 1
Reviewer 1 Report
This is an interesting and meaningful study.
- The Measures are not sufficient and lack of details. I suggest a more detailed explanation of the measures here.
- The discussion must be strengthened. For example, are the findings of this study mentioned by previous researchers? Which part of the findings is based on prior research? Or are they all new findings? What are the reasons for the author(s) to reach these findings? Or what are the possible influencing factors?
- The significance of this study needs to be more clearly stated. I suggest adding "implication" to better clarify the value of this study.
Author Response
|
Reviewer Comment |
Response |
|
Moderate English changes required |
Grammatical errors were corrected |
|
Can be improved: Include all relevant references |
We added a section "Research based on the QEWB" and provide additional literature discussing both the quality of scale as well as more results. |
|
Can be improved: Are cited references relevant to research? |
We added a section "Research based on the QEWB" and provide additional literature discussing both the quality of scale as well as more results. |
|
Can be improved: Is the research design appropriate? |
The reasearch is done. We cannot change the design ex post. |
|
Must be improved: Are the methods adequatly desribed? |
We provide a mor detailed description and analysis of the methods. We added also three sections "Research based on QWEB", "Limitations of QEWB" and "Limitations of the Study" |
|
Must be improved: Are the results clearly described? |
We completely overworked the section "Results" and provide a more preceise analysis. We also overworked the section "Discussion" and added a new section "Limitations of the study" |
|
Can be improved: Are the conclusions supported by the results? |
We overworked the section "Discussion" and added a new section "Limitations of the study" |
|
1. The Measuresd are not sufficient and lack of details. I suggest a more detailed explanation of the measures here. |
We wished the reviewer was more specific here. We added to the section "Methods" and hope we made it more sufficient. |
|
2. The discussion must be strenghtend |
We did so. |
|
3. The significance of this study needs to be more clearly stated |
We overworked the section "Discussion" and added a new section "Limitations of the study" |
Reviewer 2 Report
The authors’ work addresses important global issues related to United Nations’ SDG 3 and SDG 4 in the contexts of education for sustainable development and promotion of well-being. These are also hot topics in the agenda with recent studies reported in IJERPH and also other impactful journals. As also reflected in recent studies, there is an increasing amount of research reporting and highlighting the importance of approaching issues of happiness through eudaimonic well-being. The significance of the current work is appreciated and overall, there are some well-written parts and delivery of the manuscript. However, there are also several issues needed to be addressed. Some comments are provided below.
1.
The authors have made good reference to Waterman’s pioneer work on Questionnaire for Eudaimonic Well-Being (QEWB). There are reported attempts to capture the psychological construct of eudaimonic well-being, in particular using the widely implemented (QEWB). The background and literature review could be more comprehensive and supported by theoretical framework and recent references. IJERPH is a high quality journal with wide readership. The suggestions would be useful for providing a stronger background on how QEWB is recently applied in relevant studies of different scopes. This would also be indicative for readers’ reference on how the 6 core conceptual dimensions covered by QEWB, and to signify the need for the current study in relation to a more explicitly identified research gap and better hypothesis formulation (ln 82-88, page 2). Accordingly, relevant recent studies can be added after the description of QEWB in ln 76-78 (page 2) or in the Theory Section, after the mentioning of QEWB ln 126-127 (page 3). For example, the work by Schutte et al. (2013), which is also cited in this work (ln 180-181, page 4), can be described in more detail in the introduction or background literature review. The authors can also consider the following, or other related studies as appropriate:
Trigueros, R., Pérez-Jiménez, J. M., García-Mas, A., Aguilar-Parra, J. M., Fernandez-Batanero, J. M., Luque de la Rosa, A., ... & Navarro, N. (2021). Adaptation and Validation of the Eudaimonic Well-Being Questionnaire to the Spanish Sport Context. International Journal of Environmental Research and Public Health, 18(7), 3609.
Cara, M. A., Lobos, C., Varas, M., & Torres, O. (2022). Understanding the Association between Musical Sophistication and Well-Being in Music Students. International Journal of Environmental Research and Public Health, 19(7), 3867.
Ishii, Y., Sakakibara, R., Komoto Kubota, A., & Yamaguchi, K. (2022). Reconsidering the structure of the questionnaire for eudaimonic well-being using wide age-range Japanese adult sample: An exploratory analysis. BMC psychology, 10(1), 1-10.
2.
In view of the current work content, a section after discussion to summarize the limitation of the study is needed, for example with regards to the sampling and method constraints. The current discussion section is relatively short and should also be strengthened and incorporated with the major findings of the current work in relation to recent studies, which may also be referenced in Comment 1.
3.
More accurate and details on statistic and data manipulation should be noted. In ln 133, page 3, 37 females and 50 males are not consistent with the total number of participants in second group. Clarification or revision is needed. In ln 160 and ln 163 (page 4), the value of Z-scores displayed should be consistent.
4.
In ln 184-185, would there be any information or more descriptions on examples of activities for sustainable development that the students were engaged in?
5.
Consider to revise the keywords. I think there is some redundancy or overlapping of the current terms, so that a revision should be more indicative of all the major focus of the study.
6.
Ln 254-261, the section on Author Contributions is not completed. Only the original template information is displayed.
7.
In the abstract (ln 6 and 7, page 1), it is stated “Education for Sustainable Development (ESD, SDG 4) and human well-being (SDG 3)…”. In ln 55 and 56, a better presentation would be “ESD (SDG 4) contribute to human well-being (SDG 3)” instead of “SDG 4 and ESD contribute to SDG 3..).
8.
The key word “eudaimonic” should be checked, there are “eudaimonic” and “eudiamonic” (e.g. ln 23, page 1; ln 34, page 1; ln 61, page 2).
Author Response
| Reviewer Comment | Response |
| First of all, we want to thank the reviewer for the nice and helpful comments. It is the perhaps most helpful review we ever received. | |
| Must be improvced: Include all relevant references | We added a section "Research based on the QEWB" and provide additional literature discussing both the quality of scale as well as more results. |
| Can be improved: Is the research design appropriate? | The reasearch is done. We cannot change the design ex post. |
| Must be improved: Are the methods adequatly desribed? | We provide a more detailed description and analysis of the methods. We added also three sections "Research based on QWEB", "Limitations of QEWB" and "Limitations of the Study" |
| Must be improved: Are the results clearly described? | We provide a more detailed description and analysis of the methods. We added also three sections "Research based on QWEB", "Limitations of QEWB" and "Limitations of the Study" |
| Can be improved: Are the conclusions supported by the results? | We overworked the section "Discussion" and added a new section "Limitations of the study" |
| 1. The background and literature review could be more comprehensive and supported by recent references | We added a section "Research based on the QEWB" and provide additional literature discussing both the quality of scale as well as more results. |
| 2. A limitaions of the study section. Should be added. | We overworked the section "Discussion" and added a new section "Limitations of the study" |
| 3. More accurate and details on statistics and data manipulation should be added. | We added additional descriptive statistics that help to get a better understanding of our data. |
| 4. Would there be more information regarding examples of activities for sustainable development? | We added a description of the activities in the section "Methods: Procedure" |
| 5. Consider to revise the keywords | Keywords were revised an precised. |
| 6. Empty field "Author contribtions" | Section "Author contributions" was added. |
| 7. A better presentation would be "ESD (SDG4) contribute to human well-being (SDG3) | We did this. |
| 8. The keyword "eudaimonic" should be checked. | We did this. All spelling errors were corrected. |
Reviewer 3 Report
My congratulations this work is well constructed. The article focuses on a very interesting topic. The paper is well-written and structured. However, some issues must be improved:
- The authors should explain and discuss the reliability and validity of the questionnaire survey.
- The authors should show that there is a significant relationship between the samples (figure 1 and table p. 6)
- The authors should show that there are no significant differences in the means for individual items. (figure 1 and table p. 6)
- Table p. 6 does not have a number or title, it is necessary to complete them.
Author Response
|
Reviewer Comment |
Response |
|
Moderate English changes required |
Grammatical errors were corrected |
|
Can be improved: Include all relevant references |
We added a section "Research based on the QEWB" and provide additional literature discussing both the quality of scale as well as more results. |
|
Can be improved: Are cited references relevant to research? |
We added a section "Research based on the QEWB" and provide additional literature discussing both the quality of scale as well as more results. |
|
Can be improved: Is the research design appropriate? |
The reasearch is done. We cannot change the design ex post. |
|
Must be improved: Are the methods adequatly desribed? |
We provide a more detailed description and analysis of the methods. We added also three sections "Research based on QWEB", "Limitations of QEWB" and "Limitations of the Study" |
|
Must be improved: Are the results clearly described? |
We provide a more detailed description and analysis of the methods. We added also three sections "Research based on QWEB", "Limitations of QEWB" and "Limitations of the Study" |
|
Can be improved: Are the conclusions supported by the results? |
We overworked the section "Discussion" and added a new section "Limitations of the study" |
|
The authors should explain the reliabilty and validity of the questionn aire survey. |
We added a section "Research based on the QEWB" and provide additional literature discussing both the quality of scale as well as more results. |
|
The authors should show that there is a significant relationsship between the samples |
We provide additional statistics measures including correlations coefficient and discuss the facts of the significant relationship in the "Results" section. |
|
The authors should show that there are no significant differences in the means for individual items. |
We disagree in this point. The means of the individual items are significantly different. |
|
Table p. 6 does bnot have number and a title, it is necessary to complete them. |
We completed the description of the table. |
Round 2
Reviewer 1 Report
This paper addressed a interesting topic.The material is well organized and presented.I believe that this is a enlightening study for readers which discusses the meaningfulness of education for sustainability.
Author Response
A few grammar and spelling errors were corrected.
Reviewer 2 Report
I thank the authors for making excellent efforts in addressing the comments and their substantial work leading to further enhancement of the manuscript. In particular, I acknowledge the added references regarding the research studies of QEWB on page 3, information about the student activities and revision on page 4, and the added limitations in the last part, which help the readers achieve a more accurate and comprehensive picture of the current work.
I support the manuscript and would like to point out the following for consideration in possible future editorial work.
1.
Ln 433-435, reference items 13 and 14 correspond to the same study by Fadda et al. 2020.
2.
Ln 320, the extra “is” should be removed.
Author Response
Both comments have been corrected.